# How to Measure Adherence to a Mediterranean Diet in Dental Studies: Is a Short Adherence Screener Enough? A Comparative Analysis

**DOI:** 10.3390/nu14061300

**Published:** 2022-03-19

**Authors:** Valentin Bartha, Lea Exner, Anna-Lisa Meyer, Maryam Basrai, Daniela Schweikert, Michael Adolph, Thomas Bruckner, Christian Meller, Johan Peter Woelber, Diana Wolff

**Affiliations:** 1Department for Conservative Dentistry, University Hospital of Heidelberg, Im Neuenheimer Feld 400, 69120 Heidelberg, Germany; diana.wolff@med.uni-heidelberg.de; 2Department for Conservative Dentistry, University Hospital Tuebingen, Osianderstraße 2-8, 72076 Tübingen, Germany; exner.lea@gmx.de (L.E.); christian.meller@med.uni-tuebingen.de (C.M.); 3Institute of Nutritional Medicine, University of Hohenheim, Fruwirthstr. 12, 70599 Stuttgart, Germany; anna94.meyer@yahoo.de (A.-L.M.); m.basrai@uni-hohenheim.de (M.B.); 4Department of Nutrition Management and Nutrition Support Team, University Hospital Tuebingen, Hoppe-Seyler-Straße, 72076 Tübingen, Germany; daniela.schweikert@med.uni-tuebingen.de (D.S.); michael.adolph@med.uni-tuebingen.de (M.A.); 5Institute of Medical Biometry, Faculty of Medicine, University of Heidelberg, Im Neuenheimer Feld 130.3, 69120 Heidelberg, Germany; bruckner@imbi.uni-heidelberg.de; 6Department of Operative Dentistry and Periodontology, Faculty of Medicine, University of Freiburg, Hugstetter Str. 55, 79106 Freiburg, Germany; johan.woelber@uniklinik-freiburg.de

**Keywords:** dentistry, gingivitis, inflammation, Mediterranean diet, periodontology

## Abstract

This study aimed to evaluate the Mediterranean Diet Adherence Screener (MEDAS) in a study investigating the anti-inflammatory effect of a 6-week Mediterranean diet intervention on periodontal parameters. Data from a randomized clinical trial were analyzed for correlations between the MEDAS score and oral inflammatory parameters (bleeding on probing (BOP), gingival index (GI), and periodontal inflamed surface area (PISA)) and select nutrient intakes estimated by a food frequency questionnaire (FFQ) and a 24-h dietary recall (24dr). A mixed model, calculations of Spearman ρ, Lin’s Concordance Coefficient (CC), and Mann–Whitney U test were used for the statistical analyses. The MEDAS score was significantly negatively correlated with periodontal inflammation (BOP: CoE −0.391, *p* < 0.001; GI −0.407, *p* < 0.001; PISA −0.348, *p* = 0.001) and positively correlated with poly unsaturated fatty acids/total fat, vitamin C, and fiber intake estimates obtained from the FFQ and 24dr (ρ 0.38–0.77). The FFQ and 24dr produced heterogeneously comparable intake results for most nutrients (CC 0–0.79, Spearman ρ 0.16–0.65). Within the limitations of this study, the MEDAS was able to indicate nutritional habits associated with different levels of periodontal inflammation. Accordingly, the MEDAS can be a sufficient and useful diet screener in dental studies. Due to its correlation with oral inflammatory parameters, the MEDAS might also be useful in dental practice.

## 1. Introduction

Increasing evidence has shown that the Mediterranean diet (MedD) is associated with lower morbidity and mortality and a lower likelihood of developing certain chronic diseases [1,2,3]. The MedD is characterized by an increased consumption of vegetables, fruits, and herbs, with fatty sea fish and olive oil as the main fat sources [4]. Caries, the most common oral disease, are mainly caused by nutritional behaviors, and an increasing number of studies have indicated that malnutrition is also an important risk factor for oral inflammatory diseases such as gingivitis and periodontitis [5,6,7]. However, in dentistry, there is a lack of interventional studies on this topic; consequently, there is little evidence supporting dietary-based prevention and treatment concepts [5,8]. Thus, current periodontitis therapy guidelines require further evidence for nutritional interventions [8]. In the last decade, nutritional studies in the field of periodontology have gained increasing attention after it was found that nutrition has a beneficial effect on oral inflammation. By replacing Western dietary habits with dietary concepts found in the MedD, several nutritional interventions have shown impressive results in the reduction of oral inflammation. These dietary concepts (such as a paleolithic diet, the consumption of lettuce juice, a nitrate-rich diet, and a mainly plant-based whole foods diet) were able to reduce inflammation even though the participants had higher or constant plaque values [9,10,11,12]. Our group recently investigated the MedD in a 6-week randomized clinical trial and found comparable results to the studies above [13]. The findings of that study were in accordance with a cross-sectional study that reported lower odds of periodontitis when the participants adhered to the MedD [14]. Accordingly, the MedD offers advantages both for general and oral health and should therefore be recommended to dental patients.

However, to the best of our knowledge, how to optimally and relevantly assess the diet of dental patients with regard to the MedD remains unclear. This assessment could be in regards to diet adherence or to analyze food or nutrient intake [15,16]. When the goal is to monitor general dietary habits or adherence to recommendations, short and easy evaluable assessment screeners might be sufficient, such as the Mediterranean Diet Assessment Screener (MEDAS), which covers 14 items [16,17]. Typical food groups are queried for a minimum consumption and the result is given as a score. A score of ≥10 has been suggested to indicate sufficient adherence to the MedD [18]. MEDAS has been recently used in numerous diet studies, and negative correlations between the MEDAS score and cardiovascular diseases (CVD), symptoms of depression, odds for mobility limitations in seniors, unhealthy anthropometric parameters, occurrence of gestational diabetes, and mortality were reported [19,20,21,22,23,24]. Moreover, the MEDAS score was positively correlated with higher levels of education, nutritional knowledge, and physical activity [25,26,27]. So far, no similar study has focused on periodontal inflammatory parameters. However, the MEDAS does not allow for detailed nutrient or food analyses. The 24-h diet recall (24dr) and food frequency questionnaires (FFQs) provide the ability to analyze the consumption of certain foods or nutrient intake levels. The 24dr might demand a more intense cooperation of study participants, possibly influencing adherence and the return rate. This might lower the use of the 24dr in a dental practice compared with clinical research. The results of FFQs and the 24dr have generally been shown to agree [15,28]. The assessment of nutrient intake demands detailed documentation of one’s diet, with the consequent conversion into nutrient amounts per day prior to data analysis [29]. In studies with small sample sizes, data agreement between these methods, as well as differences between study groups, might be less valid [30]. Hence, a determination of diet adherence without the need for detailed nutrient analysis could be a beneficial alternative for those studies. Moreover, screeners such as the MEDAS might correlate positively or negatively with clinical dental parameters when there is an association between the investigated diet and oral inflammation. Consequently, in addition to the goal of low plaque levels, traditionally used in dental medicine [31,32], an increased screener score might be an additional goal of periodontal therapy. Monitoring diet adherence with an easy and efficient evaluation tool without the use of detailed nutrient analysis might facilitate study implementation and increase data validity.

Therefore, the present study aimed to determine the validity of the MEDAS using the framework and data of a recent study [13] that investigated the effect of a 6-week MedD intervention in patients with gingival inflammation. The MEDAS score was correlated with clinical data and nutritional parameters obtained from an FFQ and 24dr. The nutrient analysis results of the FFQ and 24dr were also analyzed for their correlation.

## 2. Materials and Methods

### 2.1. Study Design

The data used in the correlative analysis were from a clinical trial that investigated the effect of the MedD on gingivitis (Bartha et al., 2021). The trial was designed in accordance with the CONSORT statement for clinical trials (Figure 1). The study protocol was approved by the University of Tübingen Ethics Committee (745/2019BO2) and registered in the German Clinical Trials Register (DRKS 00025103).

### 2.2. Inclusion Criteria

Patients were included if they met all of the following criteria: had generalized gingivitis defined as >30% of sites around the teeth exhibiting bleeding on probing (BOP) [33]); ≥20 present teeth; aged 18 to 49; a BMI of 18–30 kg/m^2^; self-reported Western diet defined as a daily intake of processed carbohydrates, sugar, animal protein, saturated fatty acids, or other Western diet characteristics [34] elicited by verbal anamnesis.

### 2.3. Exclusion Criteria

Patients were excluded for any of the following reasons: periodontitis defined as a Community Periodontal Index of Treatment Needs (Ainamo et al., 1982) score > 2; smoker; severe illnesses (e.g., HIV, chronic hepatitis, cancer, illnesses of the salivary glands or the gastrointestinal tract, diabetes mellitus); pregnancy or breastfeeding; intake of antibiotics within 6 months prior to or during study; intake of anti-inflammatory drugs; treatment with medication affecting gingival bleeding; intake of probiotics; strict vegetarian, vegan, low-carb, or paleo diet; dislike or intolerance of fish, milk, or milk products; allergic to fish, fruits, or nuts; eating disorder (anorexia nervosa, bulimia, binge eating, or fasting). Those who missed more than one structured MedD training were also excluded from the study.

### 2.4. Patient Recruitment and Allocation

After recruitment through social media and institutional emails and flyers, applicants were comprehensively informed about the study and gave written consent. Eligibility screening was conducted at the Department of Conservative Dentistry, University Hospital Tübingen, Germany.

The included participants were pseudonymized and allocated to either the Mediterranean diet group (MedDG) or control group (CG) using minimalization according to gender and age (JMP, SAS Institute, Heidelberg, Germany).

### 2.5. Diet Intervention

Aiming to harmonize the clinical conditions before the start of the intervention, all participants continued their usual diet for the first 2 weeks. Furthermore, at-home dental care was equalized by asking all participants to refrain from using interdental cleaning tools and mouthwashes [11,13]. During the subsequent 6 weeks, the MedDG changed their diet to conform to a MedD. They also participated in four nutrition classes in groups of up to five participants: two sessions before starting the MedD and two meetings during the intervention period (Figure 1). Further information can be found in [13].

The Institute of Nutritional Medicine of the University of Hohenheim, Germany, provided the diet training material.

The nutrition classes were conducted by a dietician and a dentist who were specialized in the field of nutrient medicine. The first session consisted of a short lecture and group discussion on the background of the MedD and its health effects, including information on the MedD food pyramid and meal planning. Training material was distributed, and ten training tasks were set such as online research and grocery shopping; recommendations for books and apps were also given. The second session repeated much of the information and discussed selected training tasks, the MedD when eating out, canteen and restaurant menus and their agreement with the MedD, and easy preparations of MedD snacks and meals. The third session included a discussion of the first experiences with the MedD and a lecture about fat and fatty acids. Also discussed were MedD recipes and how to prepare typical Western diet meals using MedD ingredients. In the fourth session, there were further discussions about adhering to the MedD when eating out and implementation difficulties.

### 2.6. Diet Assessment and Clinical Examinations

Diet was evaluated at three time points: at week 2, near the start of the MedD intervention (*T*0); 2 weeks after *T*0 (*T*1), and at the end of the dietary intervention, 6 weeks after *T*0 (*T*2). The assessment tools used were the MEDAS, the German Health Interview and Examination Survey for Adults Food Frequency Questionnaire (DEGS-FFQ, Robert Koch Institute, Berlin, Germany), and the German Society of Nutrition 24-h dietary recall (24dr). The MEDAS is a 14-item questionnaire that queries the habitual intake of 12 typical MedD food components. The items are scored as 0 for nonadherence or 1 for adherence to the particular component [17]. A value of 10 or greater indicates MedD adherence. The DEGS-FFQ consists of 53 food items and is a reflection of one’s daily intake frequency during the previous 4 weeks. For each component, consumption frequency is answered as: one serving per month; two to three servings per month; one to two, three to four, or five to six servings per week; or one, two, three, four to five, or more than five servings per day [15]. The daily nutrient intake can then be calculated based on reference tables. For the 24dr, participants reported their food consumption and portion size at each time point.

The assessments were filled out close to each time point.

*T*0: MEDAS and DEGS-FFQ; *T*1: 24dr; *T*2: 24dr, DEGS-FFQ, and MEDAS

At *T*0 and *T*2, clinical examinations were conducted by a blinded examiner. The clinical examination used is described in Bartha et al. (2021).

### 2.7. Study Outcomes

The primary outcome was the correlation of the MEDAS score with the oral inflammatory parameters BOP, GI, and PISA. In addition, the MEDAS score was analyzed for correlations with the following nutrient intakes as evaluated by the 24dr and DEGS-FFQ: relative proportion of polyunsaturated fatty acids (PUFA) to total fat (PUFA/fat), vitamin C, and dietary fiber. The 24dr and DEGS-FFQ results at *T*2 were compared for daily intake levels of total energy (E), carbohydrates (CH), protein (P), fat (F), PUFA, saturated fatty acids (SFA), cholesterol (CHOL), glucose (GLUC), fructose (FRUC), alcohol (ALC), vitamin C (ASC), vitamin E (TOC), carotin (CAR), and fiber (FB). Additionally, for each tool, the daily nutrient intake levels were compared between the two study groups at *T*2. The results were descriptively analyzed regarding their intragroup comparison results. Changes in the MEDAS score between *T*0 and *T*2 were calculated and compared between the two groups.

### 2.8. Statistical Methods

The DEGS-FFQ data were evaluated according to the recommendations of the Robert Koch Institute. The clinical data and the nutrient analysis data obtained from the DEGS-FFQ and 24dr were analyzed using Ebis Pro (University of Hohenheim, Stuttgart, Germany). Correlations between MEDAS and clinical parameters including all timepoints were calculated using a mixed model for data with repeated measures [35]. Due to the lack of a normal distribution (Anderson–Darling test *p* < 0.05), the *T*2 intergroup comparisons were performed using the nonparametric Mann–Whitney U test. Lin´s Concordance correlation coefficient (CC) and the Spearman rank correlation between the 24dr and DEGS-FFQ was used to assess correlations between the 24dr and DEGS-FFQ. Additionally, Spearman rank correlation was used to assess correlations between the MEDAS score and nutrient intake. SAS 9.4WIN (SAS Institute, Cary, NC, USA) was used for calculation of the mixed model, Excel 16.57 (Microsoft, Redmond, WA, USA) with Real Statistics Using Excel Resource Pack for Mac, Release 8.1 (https://www.real-statistics.com, accessed on 14 March 2022) was used for calculation of CC, and JMP16.0 (SAS Institute, Cary, NC, USA) for all other statistical analyses.

## 3. Results

Of the 42 participants who met the inclusion criteria, 37 completed the study; 17 men and 20 women (Figure 2), with no difference in mean age between MedDG and CG. All participants completed the MEDAS and DEGS-FFQ at *T*0 and *T*2, except one missing MEDAS at *T*2. In total, 24 participants completed the 24dr at *T*2 (Table 1). The clinical data from our previous study are presented in Table 2. Between *T*1 and *T*2, there were statistically significant improvements in the BOP, GI, and PISA in the MedDG. The plaque values did not change in either group, while the MEDAS score significantly improved in the MedDG (Table 2). For further details see [13].

### 3.1. The MEDAS Score Was Negatively Correlated with Periodontal Inflammation and Positively Correlated with the Intake of MedD-Associated Nutrients

Analyzing all available data of MEDAS and the clinical examinations (*T*0 and *T*2), the mixed model revealed statistically significant negative correlations between the MEDAS score and BOP (correlation estimate (CoE) = −0.391, *p* < 0.001), GI (−0.407, *p* < 0.001), PISA (−0.348, *p* = 0.001), and PI (−0.23, *p* = 0.045) (Figure 3). Furthermore, using Spearman correlation analysis, the MEDAS score was positively correlated with the intake results of both detailed assessment methods (FFQ and 24dr at *T*2) for dietary fiber, vitamin C, and the relative fraction of PUFA in the daily total fat intake. The results were statistically significant for all correlations (*p* < 0.01) except 24dr vitamin C vs. MEDAS (*p* = 0.069) (Figure 4).

### 3.2. DEGS-FFQ and 24dr Nutrient Intake Results Were Heterogeneous Comparable

Calculating the concordance coefficient, the 24dr and DEGS-FFQ results showed values between 0.00 and 0.79, most pronounced for CHO (0.79). The additionally calculated nonparametric Spearman ρ displayed values from 0.16–0.65 with values less than 0.20 for CH and GLUC (Figure 5).

At *T*2, there were significant differences in daily nutrient intake levels between the two groups for several nutrients. By comparing the statistical results of each assessment method, matching significant results were found for dietary fiber and all macronutrients except carbohydrates, which was significantly higher in the CG in the 24dr analysis. Similarly, the CG cholesterol intake was significantly higher than the MedDG in the 24dr analysis. The micronutrient and alcohol intake of the MedDG was significantly higher than that of the CG in the FFQ analysis (Table 3).

## 4. Discussion

This study aimed to evaluate correlations between the MEDAS score and oral inflammatory parameters after a 6-week Mediterranean diet intervention and its comparability with the results of the DEGS-FFQ and 24dr. We found that the MEDAS score showed significant negative correlations with all assessed oral inflammatory parameters (GI, BOP, and PISA). The negative correlation was weaker, but still significant, between the MEDAS score and PI. The MEDAS score showed significant positive correlations with the intake levels of PUFA/total fat and fiber for both of the more detailed assessment methods. There were comparable values for nutrient intake as assessed by the 24dr and DEGS-FFQ.

Although the MEDAS has been evaluated for correlations with many medical and anthropometric parameters, no study has investigated its ability to assess MedD adherence within a diet intervention study in patients with gingivitis. The significant negative correlations between the MEDAS score and oral inflammatory parameters indicate the following: (i) adherence to the MedD is associated with decreased gingival inflammation and (ii) the MEDAS can be used as a dietary screening tool that can be related to the level of gingival inflammation. This tool can be recommended in research and in clinical practice, since it is easy to use by patients, and instantly interpreted by users. An explanation for the observed weak but significant correlation between the MEDAS score and PI could be that a higher adherence to a healthy diet might be associated with a higher awareness of healthy habits in general, as reported in some recent studies [25,27,36]. Alternatively, it might reflect a lower level of biofilm formation, as was seen in studies looking at sugar restriction [37,38]. In dentistry, plaque scores are traditionally used to monitor patients´ adherence to anti-inflammatory therapeutic concepts [31,32,39]. Investigating the diet in clinical studies and including dietary counseling in therapeutic concepts both require the evaluation of diet adherence. Thus, an easy evaluable form like the MEDAS might be suitable for these applications. Our results confirm the findings of Altun et al. (2021). In their cross-sectional study, they found a lower risk for periodontal disease when adherence to the MedD was increased, and they used the MEDAS as an evaluation tool [14].

The observed positive correlations between the MEDAS score and the FFQ and 24dr intake estimates for PUFA/total fat, vitamin C, and dietary fiber are comparable with the results of previous validation studies of the German and English versions of the MEDAS [17,29]. Additionally, recent studies have used the MEDAS to assess MedD adherence or adherence to healthy diets in general. In many cases, a lower MEDAS score was correlated with the presence of disease and an unhealthy lifestyle. It also correlated with the plasma metabolome profile and metabolic signature, which were able to predict the risk for CVD incidents in patients of the “Prevention with Mediterranean Diet” (PREDIMED) study [40]. In patients with depression, the occurrence of symptoms was negatively associated with the MEDAS score [23,41]. Zhao et al. found a reduced occurrence of gestational diabetes with higher MEDAS scores and with increased olive oil and pistachio consumption [24]. In summary, our results, together with those of Altun et al., are in line with previous studies that have investigated the correlations between the MEDAS score (and therefore MedD adherence) and parameters of numerous other diseases. Some studies additionally evaluated the relation between low consumption of certain food groups and lower MEDAS scores. In these studies, mostly olive oil, nuts, seafood, legumes, wine, and vegetables were the reduced food groups and the consumption of red meat and carbonated beverages was increased [23,27].

We used the DEGS-FFQ and a 24dr for the analysis of food groups and nutrient intake estimation. The nutrient and energy intake estimates from the 24dr and DEGS-FFQ at *T*2 showed significant differences for all parameters except for PUFA, SFA, ASC, and TOC. Concordance analysis revealed values ranging between 0 and 0.79. Additionally, the nonparametric Spearman correlation coefficient ranged from 0.15 to 0.65, with most values above 0.30. These results are comparable to those of validation studies of the DEGS-FFQ and other FFQs, with correlation coefficients ranging between 0.14 and 0.90 [15,28,42,43,44]. Haftenberger concluded that in cases where there are differences between the FFQ and 24dr, there is no evidence indicating which one reflects the most realistic food consumption.

In general, energy (and therefore nutrient intake) seems to be underestimated by both the 24dr and the FFQ. This underestimation has been shown in validation studies using the double-labeled water technique [45,46,47] and in comparisons of estimated sodium intake and sodium excretion in 24-h urine samples [48]. Furthermore, underestimation seems to be more pronounced with increasing BMI, especially at BMI values ≥30 [47,49]. We additionally compared the two methods regarding their ability to produce matching results when the estimated *T*2 nutrient intake was tested for statistically significant differences. Intragroup comparisons of the 24dr and DEGS-FFQ showed matching results for all macronutrients except total carbohydrates and cholesterol. Differences between the groups for micronutrients and total alcohol were shown by only one of the two assessment methods. Regarding micronutrients, all estimated differences were found to be significant in the DEGS-FFQ analysis only. This again underlines the question as to which method produces the more accurate result. As the 24dr captures the diet for only one day, the results could be susceptible to intake fluctuations, especially when it is used to display diet for a longer time period. This might also be the case if 24drs are used for more than one day. For FFQs, inaccuracies can arise because individual foods are grouped into food groups, and participants have to remember a period of 4 or more weeks. The evaluated food groups might contain foods with variable nutrient contents [50]. Thus, both tools have advantages and disadvantages. The bias resulting from the short time period covered by the 24dr could be mitigated by increasing the number of 24dr days within the study period. However, more days requires a higher degree of participant compliance, again leading to possible bias.

When monitoring diet adherence is the only goal, we find the MEDAS to be an easy to use tool for monitoring MedD adherence in a dental study. European scientific societies for periodontal diseases and caries recommend the inclusion of dietary counseling in therapeutic concepts [33]. The S3 treatment guidelines for stage I–III periodontitis suggest dietary counseling for the control of HbA1c in patients suffering from diabetes mellitus [8]. Both publications mention the need for further randomized controlled trials on this topic. Future studies should try to confirm the correlation of easy evaluation screening tools such as the MEDAS with oral inflammatory parameters to provide adherence evaluation tools for use in clinical research and clinical practice. Although the diet affects inflammation [9,10,11,12,13], the common dental therapeutic goal of low plaque values may be oversimplified because it does not address possible malnutrition, which can also lead to other nonoral diseases. Plaque scores are regularly discussed with patients and are used to motivate patients to increase oral hygiene procedures. Plaque value documentation might be supplemented with dietary scores such as the MEDAS. Like the plaque score, a dietary score could serve as a motivational approach to increase and monitor patients’ adherence to healthy diets, such as the MedD. In the context of a study with low participant numbers, an FFQ or a 24dr might be a supplement to the MEDAS, because these tools give at least an overview of the participants’ food and nutrient intake. In the current study, comparable results for the two methods were found, implicating a sufficient reflection of dietary behavior. The lower return rate of the 24dr might indicate that the FFQ was more acceptable. Future studies should evaluate whether a 24dr or an FFQ are applicable in digital forms, such as a smartphone app.

Our study had some limitations. The main limitation was the low number of participants, which decreased the statistical power of the FFQ and 24dr analysis. The reason for the low number of participants was that the main outcome parameter was the percentage of BOP; hence, the calculated number of participants was based on a predicted change in this parameter. Another limitation was that at the beginning of the study, the FFQ and 24dr had to be filled out at different time points. The 24dr was used to monitor diet adherence in the MedDG during the first 2 weeks and during the final week of the intervention. This schedule limited the number of comparable FFQs and 24drs because the tools could only be compared at the final time point. Moreover, not all participants returned or filled out their 24dr at the end of the study, leading to an even lower number of analyzable cases. This might have resulted from decreasing motivation towards the end of the study and reflected that a 24dr demands a more intense cooperation.

## 5. Conclusions

In this study, the MEDAS was sufficiently able to monitor adherence to a Mediterranean diet throughout the study period. The MEDAS score was negatively correlated with the oral inflammatory parameters BOP, GI, and PISA, but was positively correlated with nutrient intake levels as assessed by both the FFQ and 24dr. These findings indicate that in situations where diet adherence (but not nutrient intake assessment) is concerned, the MEDAS is a suitable and easy evaluation tool for use in dental practice. The MEDAS score was correlated with the observed reduction of gingival inflammation in a MedD intervention.

## Figures and Tables

**Figure 1 nutrients-14-01300-f001:**
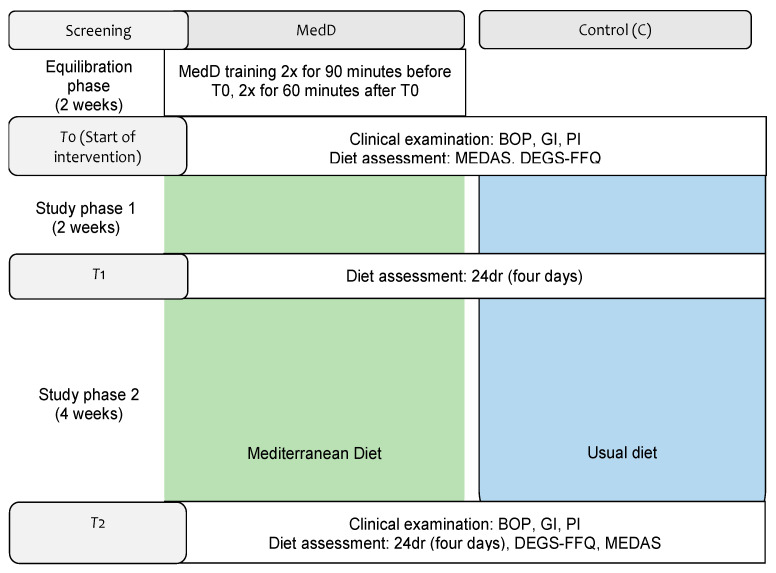
Study flow chart [13] PI = plaque index, GI = gingival index, BOP = bleeding on probing, PD = pocket depth, 24dr = 24-h dietary recall, FFQ = food frequency questionnaire, MEDAS = Mediterranean Diet Adherence Screener.

**Figure 2 nutrients-14-01300-f002:**
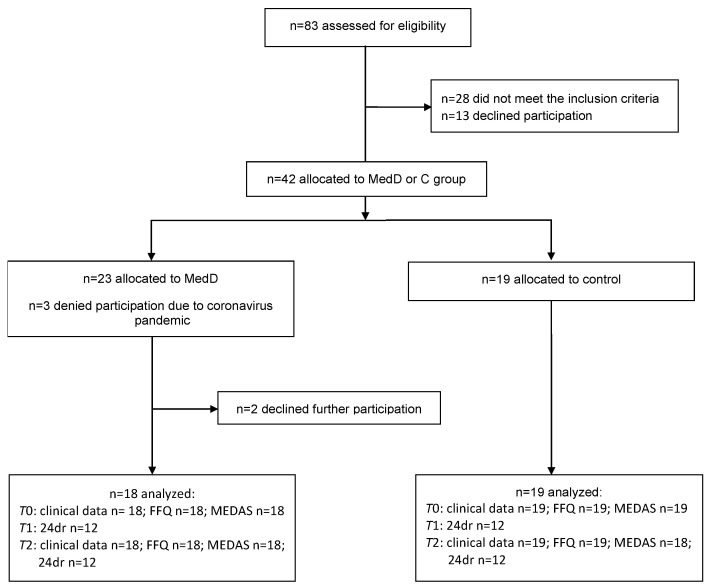
CONSORT flow diagram, modified according to [13], 24dr = 24-h dietary recall, FFQ = food frequency questionnaire, MEDAS = Mediterranean Diet Adherence Screener.

**Figure 3 nutrients-14-01300-f003:**
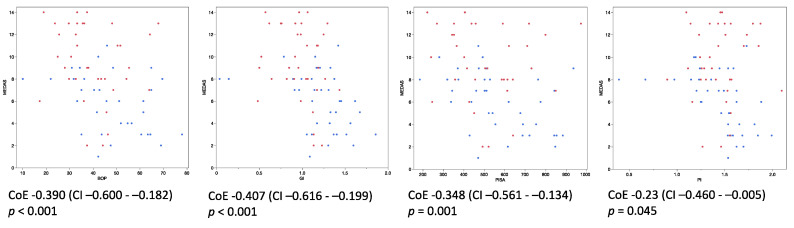
Correlation between the MEDAS score and clinical parameters (BOP, GI, PISA and PI). Scatterplots; data from *T*0 0 (blue) and *T*2 (red) for both groups (*n* = 73). MEDAS = Mediterranean Diet Adherence Screener, BOP = bleeding on probing, GI = gingival index, PISA = periodontal inflammation surface area, PI = plaque index, CoE = correlation estimate, CI = 95% confidence interval.

**Figure 4 nutrients-14-01300-f004:**
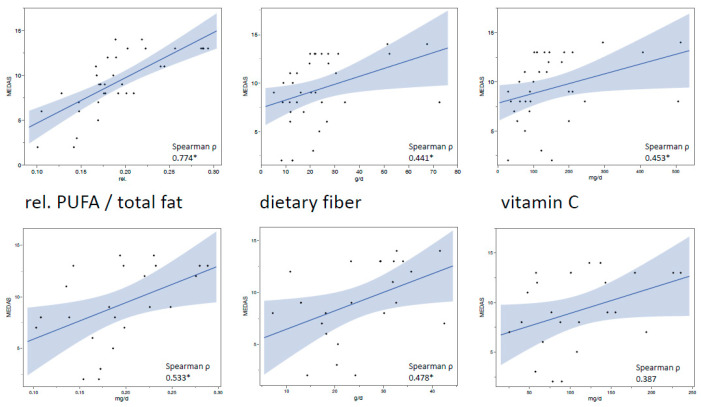
Correlations between the MEDAS score and MedD-associated nutrients. Scatterplots with regression lines and 95% confidence intervals; data from *T*2 for both groups (FFQ, *n* = 36; 24dr, *n* = 23). * *p* < 0.05, MEDAS = Mediterranean Diet Adherence Screener, PUFA = polyunsaturated fatty acids.

**Figure 5 nutrients-14-01300-f005:**
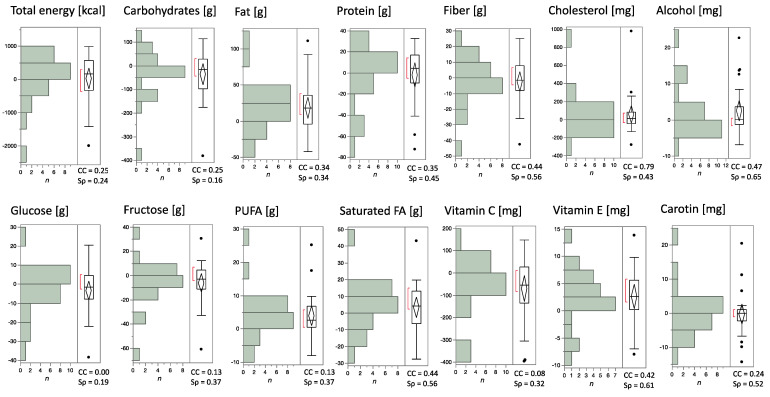
Concordances and differences between 24dr and DEGS-FFQ nutrient intake estimates for total energy, certain nutrients, and dietary fiber. Values of the calculated concordance and Spearman correlation; the data are from both groups at *T*2 (*n* = 24). PUFA = polyunsaturated fatty acids, FA = fatty acids, *n* = number, 24dr = 24 h dietary recall, FFQ = DEGS Food Frequency Questionnaire.

**Table 1 nutrients-14-01300-t001:** Baseline characteristics of MedDG and CG.

	MedDG	CG	Intergroup *p*-Value
Men	10	7	0.2536
Women	8	12	
Age (years)	32.71 ± 8.87	29.21 ± 7.17	0.198

*Note:* MedDG = Mediterranean diet group; CG = Control group. The data are from [13].

**Table 2 nutrients-14-01300-t002:** Clinical data with mean values and standard deviations, * paired *t*-test, PI = plaque index, GI = gingival index, BOP = bleeding on probing, PD = pocket depth, PISA = periodontal inflamed surface area.

	*T*1		*T*2			
	MedDG, *T*1	CG, *T*1	MedDG, *T*2	CG, *T*2	MedDG *T*1-*T*2 *Intra p-Value **	*CG T*1-*T*2 *Intra p-Value **
PI	1.51 ± 0.21	1.37 ± 0.38	1.49 ± 0.24	1.39 ± 0.24	0.560	0.823
GI	1.3 ± 0.25	1.11 ± 0.42	0.99 ± 0.22	0.97 ± 0.27	<0.001	0.093
BOP [%]	51.00 ± 14.65	43.21 ± 14.25	39.93 ± 13.74	39.74 ± 11.0	<0.001	0.151
PD [mm]	2.26 ± 0.18	2.29 ± 0.18	2.36 ± 0.17	2.36 ± 0.18	0.008	0.044
PISA [mm^2^]	616.33 ± 201.39	528.94 ± 173.48	512.02 ± 205.83	514.26 ± 148.79	0.004	0.589
MEDAS Score	5.55 ± 3.01	6.52 ± 2.17	11.89 ± 1.90	7.22 ± 2.88	<0.001	0.310

*Note*: The data are from [13].

**Table 3 nutrients-14-01300-t003:** Intergroup comparisons at *T*2 from both assessment methods. Mean values and standard deviations are given; *p*-values from the Mann–Whitney U test. Matching results of intragroup comparisons are marked (+). The Spearman rank correlation and concordance correlation coefficient (CC) is shown for the 24dr and FFQ values of both groups.

	MedD Group		C Group		24dr Inter-*p*-Value *	FFQ Inter-*p*-Value *	Matching Results Regarding Both Intragroup Comparisons	Spearman ρ: 24dr and FFQ	CC: 24dr and FFQ
	24dr (*n* = 12)	FFQ (*n* = 18)	24dr (*n* = 12)	FFQ (*n* = 19)					
Energy [kcal/d]	1647.21 ± 394.61	1769.11 ± 650.52	1963.00 ± 474.25	1785.46 ± 830.00	0.112	0.885	+	0.24	0.25 (–0.11–0.56)
Total carbohydrates [g]	160.03 ± 35.11	219.53 ± 74.74	210.39 ± 52.30	223.29 ± 146.51	0.030	0.507		0.16	0.25 (–0.03–0.49)
Total fat [g]	70.13 ± 24.65	52.83 ± 29.08	84.96 ± 44.54	64.84 ± 24.63	0.312	0.215	+	0.34	0.34 (0.01–0.60)
Total protein [g]	71.70 ± 19.90	82.06 ± 32.61	72.03 ± 18.51	65.04 ± 22.34	0.977	0.157	+	0.45	0.35 (–0.01–0.63)
Fibre [g]	29.29 ± 8.28	33.87 ± 15.24	20.40 ± 8.29	20.45 ± 17.05	0.026	0.003	+	0.56	0.44 (0.13–0.67)
Cholesterol	184.55 ± 102.15	205.10 ± 118.70	481.52 ± 609.69	341.62 ± 340.60	0.002	0.090		0.43	0.79 (0.69–0.86)
Glucose [g]	12.98 ± 9.90	18.80 ± 8.63	15.57 ± 5.31	16.36 ± 13.15	0.125	0.312	+	0.19	0.00 (–0.28–0.27)
Fructose [g]	16.38 ± 4.85	26.35 ± 12.75	19.46 ± 9.77	21.55 ± 21.73	0.624	0.126	+	0.37	0.13 (–0.15–0.38)
Polyunsatured fatty acids [g]	15.16 ± 5.65	11.37 ± 5.22	14.08 ± 9.78	9.93 ± 4.28	0.370	0.471	+	0.34	0.32 (0.02–0.56)
Satured fatty acids [g]	21.74 ± 6.77	20.32 ± 13.06	35.38 ± 17.52	29.73 ± 11.73	0.005	0.046	+	0.56	0.44 (0.07–0.70)
Total alcohol [g]	10.60 ± 7.51	8.48 ± 5.28	6.91 ± 8.65	3.84 ± 4.05	0.126	0.020		0.65	0.47 (0.16–0.69)
Vitamin C [mg]	129.79 ± 62.71	228.47 ± 123.64	89.75 ± 45.86	128.33 ± 129.74	0.126	0.009		0.32	0.08 (–0.16–0.32)
Vitamine E [mg]	15.15 ± 5.35	12.57 ± 5.93	11.50 ± 3.09	8.82 ± 5.06	0.133	0.035		0.61	0.42 (0.08–0.67)
Carotin [mg]	6.99 ± 5.17	9.37 ± 5.81	5.32 ± 6.23	3.41 ± 2.43	0.285	0.002		0.52	0.24 (–0.17–0.58)

Note: MedDG = Mediterranean diet group, CG = control group, 24dr = 24-h dietary recall, FFQ = DEGS-Food Frequency Questionnaire, CC = concordance coefficient, * by U test.

## Data Availability

The data that support the findings of this study are available from the corresponding author upon reasonable request.

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
