# Peer review of "How to Measure Adherence to a Mediterranean Diet in Dental Studies: Is a Short Adherence Screener Enough? A Comparative Analysis"

_nutrients, 2022, doi:10.3390/nu14061300_

Round 1

Reviewer 1 Report

Following, I report, point by point, my major remarks:

  • Section 3.1 line 259-260: Do the Authors mean that they included in the same analysis data from T1 e T2 all together? If so, this analysis was not totally correct. This way to compute correlation coefficient is not theoretically appropriate in a repeated-measure study, because it ignores the correlation of the two measurements within the same subjects. The appropriate analysis should account for both within-subject and across-subject variability (for example using mixed effect models).
  • Figure 5: concordance analysis would be more appropriate for the purpose. 
  • Table 2: concordance correlation coefficient would be more appropriate, instead of Sperman correlation.
  • In general, to evaluate the concordance between two tools, concordance analysis should be performed. The analysis used by the Authors may overestimate the degree of agreement between the two tools.

Some minor remarks:

  • In the abstract Authors say “The MEDAS score was significantly negatively correlated with periodontal inflammation” but p-values or C.I. were not reported.
  • In section 2.8: please specify that it is Wilcoxon signed-rank test, in order to distinguishing from the Wilcoxon rank-sum test.
  • Figure 3 and Figure 4: where are the 95% C.I.?
  • Section 3.1 line 260: correlations are statistically significant?
  • Caption of Table 2: Matching results of intragroup or intergroup?

Author Response

Dear Reviewer 1, please see the attachment.

Reviewer 2 Report

The manuscript is well written and the topic is discussed intelligently. There is one issue. In figures 1 and 2, the arrows or connecting lines are missing between the texts. They should be visible. 

Author Response

Dear Reviewer 2,

thank you for your comments.

Regarding the connecting lines and arrows: this might be a technical problem - maybe due to different operating systems (mac / windows). The lines and arrows should be visible within the uploaded pdf-version of the manuscript. We additionally revised both figures within the doc.

Round 2

Reviewer 1 Report

- Kendall´s τ (tau) is not a measure of conncordance. 

- I still can't see the 95% C.I.